# The Impact of Dietary Counseling on the Nutritional Status of Pediatric Patients with Non-IgE-Mediated Gastrointestinal Food Allergies: A Non-Randomized, Prospective Intervention Study

**DOI:** 10.3390/nu17061080

**Published:** 2025-03-19

**Authors:** Serena Coppola, Laura Carucci, Alessandra Agizza, Rita Nocerino, Rosilenia Carandente, Maria Francesca Catalano, Roberto Berni Canani

**Affiliations:** 1Department of Translational Medical Science, University of Naples “Federico II”, 80131 Naples, Italy; serena.coppola3@unina.it (S.C.); laura.carucci@unina.it (L.C.); agizza.alessandra@gmail.com (A.A.); rita.nocerino@unina.it (R.N.); rosileniacarandente@gmail.com (R.C.); mariaf.catalano97@gmail.com (M.F.C.); 2ImmunoNutritionLab, CEINGE-Advanced Biotechnologies, University of Naples “Federico II”, 80131 Naples, Italy; 3Department of Biomedicine and Prevention, University of Rome “Tor Vergata”, 00133 Rome, Italy; 4Task Force on Microbiome Studies, University of Naples “Federico II”, 80131 Naples, Italy; 5European Laboratory for the Investigation of Food-Induced Diseases, University of Naples “Federico II”, 80131 Naples, Italy

**Keywords:** malnutrition, food allergy, gastrointestinal food allergies, pediatric age, nutritional counseling, dietary counseling

## Abstract

**Background/Objectives**: Pediatric patients with non-IgE-mediated gastrointestinal food allergies (non-IgE-GIFAs) may experience alterations of nutritional status. This non-randomized, prospective intervention study investigated the impact of dietary counseling on nutritional status in pediatric patients with non-IgE-GIFAs. **Methods**: Patients of both sexes aged 0–14 years newly diagnosed with non-IgE-GIFAs received dietary counseling provided by certified pediatric dietitians immediately after diagnosis. Nutritional status parameters were assessed to identify nutritional status alterations at baseline and after 12 months of intervention (T12). **Results**: The study included 100 patients (58% male, mean age 8.5 ± 8.8 months). Non-IgE-GIFAs phenotypes included food protein-induced enteropathy (FPE, 44%), food protein-induced enterocolitis syndrome (FPIES, 11%), food protein-induced allergic proctocolitis (FPIAP, 17%), and food protein-induced motility disorders (FPIMD, 28%). At diagnosis, 1% was affected by obesity (1 FPIAP), 5% were affected by overweight (2 FPE, 1 FPIAP, and 2 FPIMD), 7% were moderately underweight (5 FPE and 2 FPIMD), 1% was severely underweight (1 FPE), 7% were moderately stunted (4 FPE, 1 FPIAP, and 2 FPIMD), 16% were moderately wasted (11 FPE, 1 FPIES, 1 FPIAP, and 3 FPIMD), and 4% were severely wasted (2 FPE and 2 FPIMD). At T12, improvements in anthropometric parameters were observed, along with a reduction in the prevalence of malnutrition by excess (6% at diagnosis vs. 2% at T12) and a reduction in the undernutrition subtypes rate, including underweight, stunting, and wasting (26% at diagnosis vs. 3% at T12, *p* < 0.001). **Conclusions**: Non-IgE-GIFAs can negatively impact the nutritional status of pediatric patients. Thus, dietary counseling could be an effective strategy for preventing and managing nutritional alterations in these patients.

## 1. Introduction

Food allergy (FA) is one of the most common chronic conditions of the pediatric age, affecting up to 10% of children in westernized countries [1]. It is estimated that up to 50% of pediatric FAs are driven by non-IgE-mediated immune mechanisms, which are primarily associated with gastrointestinal symptoms [1]. Non-IgE-mediated gastrointestinal food allergies (non-IgE-GIFAs) are conditions characterized by subacute or chronic gastrointestinal symptoms, and they are classified as food protein-induced enteropathy (FPE), food protein-induced allergic proctocolitis (FPIAP), food protein-induced enterocolitis syndrome (FPIES), and food protein-induced motility disorders (FPIMD) [2,3].

Studies reported that pediatric patients with non-IgE-GIFAs may experience nutritional status alterations [4,5,6]. The delayed onset of symptoms in non-IgE-GIFAs, combined with their overlap with other common pediatric conditions, can lead to errors and/or delays in diagnosis. As a result, chronic gastrointestinal symptoms caused by persistent exposure to food allergens may exacerbate nutrient losses due to intestinal malabsorption, thereby negatively affecting growth and nutritional status [2,3].

Upon diagnosis, as highlighted in most FA management guidelines, patients with non-IgE-GIFAs should receive proper dietary guidance to prevent inappropriate elimination diets and to ensure the adequate replacement of excluded foods [7,8]. This approach is essential to minimize the risk of micronutrient deficiencies, anemia, rickets, failure to thrive, and kwashiorkor, particularly in cases of multiple FAs, where the risk may rise as the number of foods avoided increases [6,9,10,11]. Therefore, preventing malnutrition is crucial, as its medium- and long-term effects include impaired growth, delayed cognitive development, weakened immunity, increased susceptibility to infections, and long-term health complications [12].

To date, the effects of dietary counseling on the nutritional status of pediatric patients with non-IgE-GIFAs are still poorly explored. Most studies focus on patients with IgE-mediated allergies [9,10,11,13], but due to the high prevalence of non-IgE-mediated forms, it is essential to explore this area further. Hence, this study was designed to evaluate the impact of dietary counseling on the nutritional status of pediatric patients with non-IgE-GIFAs after 12 months of nutritional intervention.

## 2. Materials and Methods

### 2.1. Study Design and Ethics

This was a pilot, monocentric, non-randomized, prospective intervention study conducted from January 2017 to January 2020 at the Tertiary Center for Pediatric Allergy, Gastroenterology, and Nutrition at the Department of Translational Medical Science of the University of Naples Federico II, Naples (Italy). The study was approved by the local ethics committee and was conducted in accordance with the Helsinki Declaration (Finland revision, 2024), the standards of Good Clinical Practice (CPMP/ICH/135/95), and the pertinent European and Italian regulations on data protection. This study is a part of the “Non-IgE-mediated Food Allergies in Children” (NIGEFA) project, which was registered on https://clinicaltrials.gov/ (accessed on 1 February 2025) with the identifier NCT04462978, a prospective study assessing the pathophysiological mechanisms, natural history, and disease course of non-IgE-GIFAs [14].

### 2.2. Participants

Consecutive pediatric patients, newly diagnosed with non-IgE-GIFAs according to the current diagnostic criteria [2,15], of both sexes and aged 0–14 years were eligible for the study. The exclusion criteria were the following: lack of written informed consent; age >14 years; infectious diseases; chronic systemic diseases; malignancies; immunodeficiencies; autoimmune diseases; celiac disease; metabolic and genetic diseases; cystic fibrosis or other forms of primary pancreatic insufficiency; malformations or previous major surgery procedures of the gastrointestinal, cardiovascular, urinary, or respiratory tract; psychiatric and neurological diseases; and eosinophilic gastrointestinal disorders.

### 2.3. Study Outcomes

The main outcome of this study was to investigate the impact of dietary counseling on the nutritional status of pediatric patients with non-IgE-GIFAs by assessing anthropometric parameters and relative z-scores to identify the rate of nutritional status alterations (overweight, obesity, underweight, stunting, and wasting) after 12 months of intervention. Other outcomes included assessing whether the rate of nutritional status alterations was influenced by the presence of a single FA versus multiple FAs or by the diagnostic delay (measured as the number of days between the onset of symptoms and the diagnosis).

### 2.4. Sample Size

As this is a pilot study, a formal sample size calculation was not performed. Instead, we consecutively enrolled the first 100 pediatric patients with a confirmed diagnosis of non-IgE-GIFAs. This approach aligns with similar studies investigating nutritional status alterations in pediatric patients with FAs, where comparable sample sizes have provided meaningful insights into growth patterns and dietary impacts [9,11,13,16]. Given the exploratory nature of this study and the relatively rare condition under investigation, our sample size was deemed appropriate for detecting clinically relevant trends in nutritional status changes following dietary counseling. Moreover, the findings from this study will serve as a foundation for future research with larger, statistically powered cohorts.

### 2.5. Study Procedures

At baseline (T0), soon after the diagnosis of non-IgE-GIFAs, written informed consent from the parents/legal guardian of each patient and a written assent form from patients of 5 years and older were obtained, and the enrolled subjects underwent a complete anamnestic and clinical evaluation by an experienced multidisciplinary team composed of pediatric allergists, pediatric gastroenterologists, and certified dietitians. Anamnestic and demographic data (i.e., sex, age at diagnosis and at symptom onset, familial history of allergic diseases, delivery mode, breastfeeding, mono- or polyallergic sensitization, days of diagnostic delay, etc.) were recorded in a dedicated clinical chart. Furthermore, patients underwent a full clinical examination, including the collection of weight, length/height, and relative z-scores (weight-for-age, length/height-for-age, weight-to-length ratio, and body mass index), and personalized dietary counseling was provided.

One month after enrollment (T1), clinical and nutritional status parameters were reassessed, and adherence to the prescribed elimination diet was monitored through a 24 h recall. After 6 months (T6) and 12 months (T12) from the diagnosis, a full anamnestic and clinical evaluation was performed, nutritional status parameters were collected, adherence to the prescribed elimination diet was monitored through a 24 h recall, and allergy screening tests and oral food challenges (OFC) were performed to explore tolerance acquisition in each study subject. For patients affected by FPIES, the OFC was scheduled annually (12 months after diagnosis), in accordance with current guidelines [1].

Unscheduled visits were performed as needed, either in response to allergic symptoms or the development of other FAs that required adjustments to dietary therapy.

### 2.6. Parameters of Nutritional Status

Anthropometric measurements were performed by experienced dietitians after a 12 h fast at baseline (T0) and after 1, 6, and 12 months of follow-up (T1, T6, and T12). Body weight and length or height were taken by participants without or in light clothing and without shoes using a mechanical scale and a stadiometer or altimeter, following standard procedures [17]. In patients aged >2 years, body mass index (BMI) was calculated by relating weight and height squared (kg/m^2^). The relative anthropometric parameter z-scores were derived using the World Health Organization (WHO) software Anthro version number 3.2.2 and were used to interpret growth measurements. Using the WHO growth charts [18], nutritional status alterations were classified as overweight (weight-for-length or BMI > 2 and ≤ 3 z-score up to 5 years of age; BMI > 1 and ≤ 2 z-score from 5 to 19 years of age), obesity (weight-for-length or BMI > 3 z-score up to 5 years of age; BMI > 2 z-score from 5 to 19 years of age), moderately underweight (weight-for-age ≤ −2 and ≥ −3 z-score), severely underweight (weight-for-age < −3 z-score), moderate stunting (length/height-for-age ≤ −2 and ≥ −3 z-score), severe stunting (length/height-for-age < −3 z-score), moderate wasting (weight-for-length or BMI ≤ −2 and ≥ −3 z-score), and severe wasting (weight-for-length or BMI < −3 z-score).

### 2.7. Dietary Counseling

At T0, soon after the non-IgE-GIFAs diagnosis, all enrolled patients underwent personalized dietary counseling by experienced certified dietitians as part of the standard management of patients with FAs, following nutritional protocol that is regularly used at our center for the management of pediatric patients affected by FAs. The protocol is based on the Position Statements provided by the Italian Society for Pediatric Gastroenterology, Hepatology, and Nutrition (SIGENP) and the Italian Society for Pediatric Allergy and Immunology (SIAIP) [3], as well as by the Italian Society of Pediatric Nutrition and the Italian Society of Pediatric Allergy and Immunology [19]. It has also been previously adopted by our group in a former study [17].

Dietary counseling began with the implementation of an elimination diet tailored to exclude the allergen(s) responsible for non-IgE-GIFAs. The primary goal of this diet was to prevent allergic reactions and the persistence of symptoms caused by accidental allergen(s) exposure. Equally important was the need to prevent nutritional deficiencies and to ensure proper growth and development of patients. Patients and their parents/legal guardians were carefully educated on topics such as contamination risks and cross-reactions, so information on safe food preparation at home and information on safely eating outside of the home environment were provided. Families were instructed on label reading and on the terminology used on labels of foods and of non-food products (i.e., drugs, cosmetics, and supplements), so lists of commercially available foods free from the allergen responsible for the allergy were provided, along with recipes for homemade preparations that do not contain the responsible allergen(s). The dietitians provided guidance on replacing allergenic foods with alternative nutrient sources to ensure the daily recommended intake for children, tailored to their age and sex [20]. For instance, in non-breastfed infants with a cow’s milk allergy, the use of specialized infant formulas was recommended [6]. When necessary, if the nutritional requirements for certain nutrients could not be met through foods alone, oral supplementation was prescribed (e.g., omega-3, calcium, vitamin D, etc.). Finally, it was recommended to follow a healthy dietary pattern, such as the Mediterranean diet, whose anti-inflammatory and anti-allergic effects are well known, and the principles to follow were thoroughly explained [21]. In turn, the intake of ultra-processed foods, which are rich in harmful compounds such as advanced glycation end products and have been linked to an increased risk of FAs, was strongly discouraged [22,23,24].

During the follow-up visits T1 and T6, if patients developed other FAs that required adjustments to diet therapy, the dietary counseling was again carried out.

### 2.8. Data Management and Statistical Analysis

A clinical trial monitor verified the completeness, clarity, consistency, and accuracy of the clinical forms. All data were collected anonymously and entered into the study database using a single-entry method. The database underwent standard data cleaning procedures and was locked before statistical analysis by the statistical team. The Kolmogorov–Smirnov test was applied to assess the normality of continuous variables, which were presented as mean (SD) when normally distributed. Categorical variables were expressed as counts and percentages, with the χ^2^ test and Fisher’s exact test applied when appropriate. Comparisons of data between enrollment and 12 months from baseline were conducted using the paired Student’s *t*-test. A two-sided *p*-value of <0.05 was considered statistically significant. All statistical analyses were performed using SPSS for Windows (version 23.0, SPSS Inc., Chicago, IL, USA).

## 3. Results

### 3.1. Study Population

In the study, the first 100 consecutive pediatric patients newly diagnosed with non-IgE-GIFAs who were referred to the Tertiary Center were enrolled. The main anamnestic and clinical features of the study population are reported in Table 1.

Patients enrolled were affected by the following non-IgE-GIFAs manifestations: FPE in 44% of cases, FPIES in 11%, FPIAP in 17%, and FPIMD in 28%. The majority (80%) had a cow’s milk protein allergy, followed by egg (26%), soy (17%), wheat (14%), meat (13%), rice (11%), legumes (10%), fish (8%), and other allergens (9%).

All patients completed the study, and none were lost in the follow-up.

### 3.2. Improvement of Anthropometric Parameters After 12 Months of Intervention

Nutritional parameters of the enrolled patients, categorized by type of non-IgE-GIFAs (FPE, FPIES, FPIAP, or FPIMD), were assessed before and after the intervention, as shown in Table 2.

At baseline, a significant difference was observed between the body length/height-for-age z-scores of FPE vs. FPIAP patients (*p* = 0.013).

After the intervention, significant improvements were reported across various nutritional parameters:-In the FPE group, there were improvements in body weight, the body weight-for-age z-scores, length/height, the weight-to-length ratios, and the BMI z-scores (*p* < 0.0001);-In the FPIES group, there were improvements in body weight and length/height (*p* < 0.0001);-In the FPIAP group, there were improvements in body weight (*p* = 0.006) and length/height (*p* = 0.003);-In the FPIMD group, there were improvements in body weight, length/height (*p* < 0.0001), and the weight-to-length ratios or BMI z-scores (*p* = 0.01).

Overall, at T12, the entire study population (*n* = 100) showed significant improvements in body weight, body length/height, the body weight-for-age z-scores, and the weight-to-length ratios or BMI z-scores (*p* < 0.001).

Additionally, significant differences were observed at T12 in the body weight-for-age z-scores (*p* = 0.032) and body length/height-for-age z-scores (*p* = 0.025) between FPE and FPIAP patients.

### 3.3. Reduction in Nutritional Status Alterations After 12 Months of Intervention

In Table 3, the rate of nutritional status alterations at baseline and after 12 months of follow-up in the enrolled patients are reported, divided by type of non-IgE-GIFAs manifestations (FPE, FPIES, FPIAP, or FPIMD) and taken all together, according to the WHO classification. No significant differences among groups at baseline and at T12 in the nutritional status alteration rate were observed.

A reduction in the rate of nutritional status alterations after 12 months of intervention was observed, and, in particular, a significant reduction in the rate of FPE patients affected by moderate wasting (25% vs. 0%, *p* < 0.0001). Furthermore, a reduction in the prevalence of malnutrition by excess (obesity and overweight) (6% at diagnosis vs. 2% at T12) was observed, and a significant reduction in the rate of undernutrition subtypes, including underweight, stunting, and wasting (26% at diagnosis vs. 3% at T12, *p* < 0.001), was observed.

The final response rate, defined as the percentage of patients who showed the resolution of at least one nutritional status alteration at T12, was 90.3%.

Finally, we did not observe that polysensitization (multiple FAs), nor the days of diagnostic delay (expressed as days elapsed between the onset of symptoms and diagnosis), had an impact on nutritional status alterations (*p* > 0.05).

## 4. Discussion

Pediatric patients with non-IgE-GIFAs may experience alterations in their nutritional status, including poor growth, micronutrient deficiencies, and feeding difficulties [9,10,11,25]. The most commonly responsible antigens for FAs (cow’s milk, eggs, wheat, fish, etc.) are those with high nutritional value, as they are sources of essential nutrients such as proteins and micronutrients, which are crucial for the proper growth and development of children [26]. Unsupervised elimination of these foods can have an impact on nutrient intake, as reported in previous observational studies [9,10,27,28,29,30,31]. Inadequate specific nutrient intake can negatively impact the nutritional status of FA pediatric patients. For instance, several studies reported nutritional status alterations in pediatric patients affected by FAs [25]. However, comparing data across different studies is challenging due to variations in the diagnostic criteria and classification of FAs, as well as differences in the timing and setting of data collection (e.g., at diagnosis vs. after dietary follow-up; Primary-, Secondary-, or Tertiary-care Centers) and the growth reference curves used.

In the present study, we assessed the rate of nutritional status alterations by evaluating the anthropometric parameters and relative z-scores of non-IgE-GIFAs pediatric patients at diagnosis, and we evaluated the impact of dietary counseling on their nutritional status after 12 months of intervention.

We observed an almost similar rate of underweight status and stunting at diagnosis, affecting 8% and 7% of our patients, as those reported in a study yielding 97 patients with confirmed IgE- and/or non-IgE-mediated FAs from 13 different centers throughout the UK (8.5% and 11.1%). On the contrary, we reported a higher rate of wasting: 20% in our study vs. 3.7% observed in the UK study [16]. Even though the authors applied the same WHO standards we used for malnutrition classification, their survey did not specify whether the collected data referred to the time of diagnosis. As a result, it remains unclear whether the subjects were newly diagnosed food-allergic children or those already following an elimination diet, which could significantly impact the growth data. Moreover, patient data were gathered across various healthcare settings, including two Primary, eight Secondary, and three Tertiary Centers. Given these differences, we do not consider it appropriate to compare their findings with ours, which were obtained both at diagnosis and after a structured dietary intervention exclusively in a Tertiary Center, and specifically in patients with non-IgE-mediated allergies.

A lower rate of being underweight and stunting and a higher rate of wasting were observed in our cohort at diagnosis, compared to the non-IgE-mediated allergic Brazilian cohort described by Vieira MC. et al. [32]. Although nutritional status was evaluated in patients with non-IgE-mediated FAs seen at first evaluation, the authors applied growth chart references different from ours, and data were collected in pediatric gastroenterology offices, which hinders an accurate data comparison.

After 12 months of intervention, our study observed a slightly different pattern of nutritional status alterations compared to the findings of Meyer R. et al., who assessed the nutritional status of non-IgE-GIFA patients at a Tertiary Gastroenterology Department after four weeks on an elimination diet. Specifically, our cohort exhibited a lower prevalence of stunting at T12 (3% vs. 9% in the Meyer R. et al. study), a slightly lower prevalence of wasting (0% vs. 2.2%), and a similar rate of being overweight (1% vs. 2.2%) [33]. Despite using the same WHO reference curves and the evaluation of the same allergy type, the differences in results are likely due to the fact that all patients in the Meyer R. et al. study received individualized dietetic advice for an elimination diet lasting only 4 weeks, significantly shorter than our 12-month duration. This shorter timeframe was likely insufficient for a meaningful data comparison.

Finally, our cohort at follow-up exhibited lower patterns of being underweight, stunting, wasting, and being overweight compared to the pooled data from 430 pediatric patients with IgE- and non-IgE-mediated FAs across twelve allergy centers (1% vs. 6% in the Meyer R. et al. study; 3% vs. 9%; 0% vs. 5%; and 1% vs. 3%, respectively) [34]. Comparing our data with theirs is not entirely accurate for several reasons. First, not all patients across the 12 centers came from Tertiary Centers or had access to nutritional counseling for the elimination diet due to dietitian/nutritionist unavailability. Second, the duration of the elimination diet among the analyzed patients was neither known nor standardized. Third, patients with both IgE-mediated and non-IgE-mediated forms of allergies were included in the study. Lastly, since the study does not encompass all European populations, drawing direct comparisons with our data remains challenging.

Our analysis revealed that the number of FAs (i.e., the presence of polysensitization) did not negatively affect the rate of nutritional status alterations. Our findings are in disagreement with other studies, which reported that growth tends to be impaired as the number of food allergens avoided increases [13,16,34,35,36,37]. However, others have reported findings consistent with ours [38]. Inconsistencies in the data may stem from the fact that most studies report correlations between the number of eliminated allergens and weight or length/height-for-age z-scores. In some cases, these correlations were observed even when z-scores remained within normal ranges, rather than directly evaluating whether individuals with multiple allergies have a higher prevalence of nutritional status alterations [34,36,37,38].

In our study, diagnostic delay was not a factor that influenced alterations in nutritional status. This notion seems speculative, as there is no existing evidence to support it, and our findings do not corroborate this hypothesis.

To our knowledge, this is the first study aimed at assessing nutritional status alterations in pediatric patients with non-IgE-mediated allergies, classified according to the different forms of the disease. This represents a novel approach, as although other studies have examined growth parameters in patients with non-IgE-mediated allergies, none have analyzed nutritional status alterations across the various forms of these allergies [16,32,33,34]. In our cohort, the highest rates of undernutrition subtypes at diagnosis, including being underweight, stunting, and wasting, were found in the FPE subpopulation. We feel that patients with FPE may experience more pronounced alterations in their nutritional status for several reasons. Inflammation and mucosal damage resulting from allergic reactions to food antigens can interfere with proper nutrient absorption in the gastrointestinal tract [39,40]. Such damage impairs the intestinal mucosal barrier, leading to increased intestinal permeability [41,42,43]. Chronic gastrointestinal symptoms, such as diarrhea, vomiting, and abdominal pain, can further contribute to poor nutrient intake and loss, exacerbating nutritional deficiencies [44,45]. Moreover, the restrictive nature of the elimination diet typically used to manage the condition can limit food choices and result in insufficient intake of vital nutrients, especially if not properly monitored by specialists [44].

In this study, we demonstrated that dietary counseling is an effective strategy for rapidly correcting and preventing nutritional imbalances in FA pediatric patients, in line with previous findings from our group and others [17,38].

## 5. Conclusions

Our study has several strengths. One of the key aspects is that it was conducted by a multidisciplinary team in a Tertiary Center and involved patients who received a confirmed diagnosis of non-IgE-GIFAs. We assessed different subtypes of nutritional status alterations, categorizing them into moderate and severe forms, while also evaluating overnutrition in patients classified according to the different forms of non-IgE-mediated FAs. Another advantage is that our study is reproducible, as the dietary intervention key steps were standardized.

The limitations of our study include the lack of analysis of body composition and biochemical parameters of nutritional status and the absence of dietary nutrient intake assessment through food diaries. Furthermore, we did not explore the underlying causes of nutritional status alterations, such as barrier permeability or inflammatory markers, among others. Finally, since this is a non-randomized, single-center study with a relatively small sample size, our results cannot be directly generalized to larger populations or compared with centers that adopt different nutritional approaches. However, our Tertiary Reference Center receives patients referred from other hospitals and family pediatricians across the country, making our study population reasonably representative of individuals affected by these conditions.

## Figures and Tables

**Table 1 nutrients-17-01080-t001:** Anamnestic and clinical features of the study population.

	Study Population (*n* = 100)
Male, *n* (%)	58 (58)
Spontaneous delivery, *n* (%)	39 (39)
Born at term, *n* (%)	94 (94)
Breastfed at diagnosis, *n* (%)	6 (6)
Familial allergy risk, *n* (%)	64 (64)
Age at symptom onset, mean (±SD)	4.9 (5.4)
Age at diagnosis, mean (±SD)	8.5 (8.8)
Monosensitized, *n* (%)	53 (53)
Diagnostic delay days, median (IQR)	60 (150)

SD, standard deviations; IQR, interquartile range.

**Table 2 nutrients-17-01080-t002:** Anthropometric parameters of the study population at baseline and after 12 months of intervention.

	FPE	FPIES	FPIAP	FPIMD	All Patients
N.	44	11	17	28	100
Body weight, mean kg (±SD)					
at baseline	10.5 (4.5)	10.8 (3.7)	9.3 (3.6)	10.8 (4.7)	10.4 (4.3)
after 12 months	12.8 (3.9) *	12.5 (3.4) *	11.9 (4.5) *	12.8 (4) *	12.6 (4) *
Body length/height, mean cm (±SD)					
at baseline	80.1 (15.9)	80.6 (13.3)	74.2 (11.9)	80.2 (15.4)	79.2 (14.8)
after 12 months	88.2 (13.2) *	88.2 (10.9) *	83.1 (14.9) *	87.5 (13.2) *	87.1 (13.2) *
BMI, mean (±SD)					
at baseline	15.4 (1.5)	16.2 (0.3)	16 (-)	16.4 (1.4)	15.9 (1.4)
after 12 months	15.7 (1.1)	16.4 (1.1)	17.1 (2.3)	18.8 (7.8)	16.9 (4.5)
Body weight-for-age z-score, (±SD)					
at baseline	−0.49 (1.2)	0.05 (0.8)	0.16 (1.3)	−0.12 (1.1)	−0.22 (1.1)
after 12 months	−0.09 (1) *^,#^	0.13 (0.6)	0.49 (0.8) ^#^	0.16 (0.9)	0.1 (0.9) *
Body length/height-for-age z-score, (±SD)					
at baseline	−0.12 (1.3) ^#^	0.32 (1.2)	0.89 (1.6) ^#^	0.21 (1.3)	0.19 (1.3)
after 12 months	−0.17 (1.4) ^#^	0.42 (1.1)	0.65 (0.92) ^#^	0.14 (1.2)	0.12 (1.2)
Weight-to-length ratio or BMI z-score, (±SD)					
at baseline	−0.67 (1.5)	−0.05 (1)	−0.18 (1.4)	−0.36 (1.7)	−0.43 (1.5)
after 12 months	0.12 (0.9) *	−0.003 (0.9)	0.40 (0.96)	0.25 (1) *	0.19 (0.9) *

SD, standard deviations; BMI, body mass index. ^#^ *p* < 0.05, FPE vs. FPIAP. * *p* < 0.05, 12 months vs. baseline.

**Table 3 nutrients-17-01080-t003:** Nutritional status alteration rate of the study population at baseline and after 12 months of intervention according to the WHO classification.

	FPE	FPIES	FPIAP	FPIMD	All Patients
N.	44	11	17	28	100
Obesity, *n* (%)					
at baseline	0	0	1 (5.9)	0	1 (1)
after 12 months	0	0	1 (5.9)	0	1 (1)
Overweight, *n* (%)					
at baseline	2 (4.5)	0	1 (5.9)	2 (7.1)	5 (5)
after 12 months	1 (2.3)	0	0	0	1 (1)
Moderate underweight, *n* (%)					
at baseline	5 (11.4)	0	0	2 (7.1)	7 (7)
after 12 months	1 (2.3)	0	0	0	1 (1)
Severe underweight, *n* (%)					
at baseline	1 (2.3)	0	0	0	1 (1)
after 12 months	0	0	0	0	0
Moderately stunted, *n* (%)					
at baseline	4 (9.1)	0	1 (5.9)	2 (7.1)	7 (7)
after 12 months	3 (6.8)	0	0	0	3 (3)
Severely stunted, *n* (%)					
at baseline	0	0	0	0	0
after 12 months	0	0	0	0	0
Moderately wasted, *n* (%)					
at baseline	11 (25)	1 (9.1)	1 (5.9)	3 (10.7)	16 (16)
after 12 months	0 *	0	0	0	0
Severely wasted, *n* (%)					
at baseline	2 (4.5)	0	0	2 (7.1)	4 (4)
after 12 months	0	0	0	0	0

* At baseline vs. after 12 months, *p* < 0.05.

## Data Availability

Data generated or analyzed during this study can be found within the article.

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
