# Peer review of "The Impact of Dietary Counseling on the Nutritional Status of Pediatric Patients with Non-IgE-Mediated Gastrointestinal Food Allergies: A Non-Randomized, Prospective Intervention Study"

_nutrients, 2025, doi:10.3390/nu17061080_

Round 1

Reviewer 1 Report

Comments and Suggestions for Authors

The study presents a relevant and timely investigation into the impact of dietary counseling on pediatric patients with non-IgE-mediated gastrointestinal food allergies (non-IgE-GIFAs). It is well-structured and follows a logical flow, addressing an important gap in research regarding the nutritional management of these patients. However, there are some areas that could be improved for clarity, methodological rigor, and completeness of discussion

  • The lack of randomization may introduce selection bias. If randomization was impractical, a matched control group could strengthen findings. Being monocentric, the results may not generalize to broader populations. Consider acknowledging this as a limitation and discussing potential differences in dietary counseling approaches across centers.
  • Differences in diagnostic criteria, classification of food allergies, growth reference curves, and the timing of assessments complicate direct comparisons. The study acknowledges these limitations but could benefit from further elaboration.
  • What specific nutritional guidelines were followed? Were standardized nutrition protocols (e.g., ESPGHAN) used? Was compliance with the dietary plan assessed through food diaries or nutrient intake analysis?
  • Given the longitudinal nature, repeated measures ANOVA or mixed-effects models might better capture intra-individual changes. Were adjustments made for multiple comparisons?
  • How were missing values treated? Imputation methods (e.g., LOCF, multiple imputation) should be clarified.

Despite its limitations and the previously mentioned issues, this study represents a significant step forward in improving nutritional treatment techniques for paediatric patients with non-IgE-mediated gastrointestinal food allergies.

Author Response

The study presents a relevant and timely investigation into the impact of dietary counseling on pediatric patients with non-IgE-mediated gastrointestinal food allergies (non-IgE-GIFAs). It is well-structured and follows a logical flow, addressing an important gap in research regarding the nutritional management of these patients. However, there are some areas that could be improved for clarity, methodological rigor, and completeness of discussion

The lack of randomization may introduce selection bias. If randomization was impractical, a matched control group could strengthen findings. Being monocentric, the results may not generalize to broader populations. Consider acknowledging this as a limitation and discussing potential differences in dietary counseling approaches across centers.

We included in the study 100 pediatric patients consecutively observed at our Tertiary Center because of non-IgE-mediated gastrointestinal food allergies. The Pediatric Allergy Gastroenterology and Nutrition Comprehensive Education, Treatment and Support Program at the University of Naples Federico II is one of the largest providers of pediatric allergy and gastroenterology services in the EU. The Program is designed to treat all aspects of food allergy and other pediatric allergic diseases, including medical, dietary, social, and psychological concerns. Our Tertiary Reference Center welcomes patients from other hospitals and/or family pediatricians from all regions of the Country. Thus, even though these patients were evaluated at just one Center, we feel that this population could be well representative of patients affected by these conditions. This point is also supported by the fact that our baseline data are well in line with those reported by other Authors operating in different parts of the World (e.g., Meyer, R.; De Koker, C.; Dziubak, R.; Venter, C.; Dominguez-Ortega, G.; Cutts, R.; Yerlett, N.; Skrapak, A.K.; Fox, A.T.; Shah, N. Malnutrition in children with food allergies in the UK. J Hum Nutr Diet. 2014, 27, 227-35.).

As pointed out by Reviewer 1, randomizing patients into two groups (intervention vs. control groups) was not feasible because nutritional intervention is an essential part of the standard care of patients with food allergies, making it ethically and practically impossible to exclude it.

These two limitations have been recognized in the revised version of the Discussion Section of the paper.

Differences in diagnostic criteria, classification of food allergies, growth reference curves, and the timing of assessments complicate direct comparisons. The study acknowledges these limitations but could benefit from further elaboration.

The text has been revised according to this suggestion.

What specific nutritional guidelines were followed? Were standardized nutrition protocols (e.g., ESPGHAN) used? Was compliance with the dietary plan assessed through food diaries or nutrient intake analysis?

For the study, we adopted the nutritional protocol that is regularly used at our Center for the management of pediatric patients affected by food allergy. The protocol is based on the Position Statements provided by the Italian Society for Pedi-atric Gastroenterology, Hepatology, and Nutrition (SIGENP) and the Italian Society for Pediatric Allergy and Immunology (SIAIP) (Berni Canani R, Caffarelli C, Calvani M, Martelli A, Carucci L, Cozzolino T, Alvisi P, Agostoni C, Lionetti P, Marseglia GL. Diagnostic therapeutic care pathway for pediatric food allergies and intolerances in Italy: a joint position paper by the Italian Society for Pediatric Gastroenterology Hepatology and Nutrition (SIGENP) and the Italian Society for Pediatric Allergy and Immunology (SIAIP). Ital J Pediatr. 2022 Jun 10;48(1):87. doi: 10.1186/s13052-022-01277-8. PMID: 35689252; PMCID: PMC9188074.), as well as by the Italian Society of Pediatric Nutrition and the Italian Society of Pediatric Allergy and Immunology (Giovannini M, D'Auria E, Caffarelli C, Verduci E, Barberi S, Indinnimeo L, Iacono ID, Martelli A, Riva E, Bernardini R. Nutritional management and follow up of infants and children with food allergy: Italian Society of Pediatric Nutrition/Italian Society of Pediatric Allergy and Immunology Task Force Position Statement. Ital J Pediatr. 2014 Jan 3;40:1. doi: 10.1186/1824-7288-40-1. PMID: 24386882; PMCID: PMC3914356.). It has also been previously adopted by our group in a former study (Berni Canani R, Leone L, D'Auria E, Riva E, Nocerino R, Ruotolo S, Terrin G, Cosenza L, Di Costanzo M, Passariello A, Coruzzo A, Agostoni C, Giovannini M, Troncone R. The effects of dietary counseling on children with food allergy: a prospective, multicenter intervention study. J Acad Nutr Diet. 2014 Sep;114(9):1432-9. doi: 10.1016/j.jand.2014.03.018. Epub 2014 Jun 3. PMID: 24933388.).

The adherence to the dietary plan was monitored through an interview with the parents/tutors of the patients. During the interview we monitored clinical and nutritional parameters, and the diet was evaluated through a 24-h recall for monitoring the adherence to the prescribed elimination diet and nutrients intake.

Given the longitudinal nature, repeated measures ANOVA or mixed-effects models might better capture intra-individual changes. Were adjustments made for multiple comparisons?

In our analysis, we opted for Student’s t-test to compare anthropometric parameters between baseline (T0) and the 12-month follow-up (T12), given that our primary objective was to assess the impact of dietary counseling on nutritional status over two time points. However, we have added more information in the revised version of Result section of the paper.

How were missing values treated? Imputation methods (e.g., LOCF, multiple imputation) should be clarified.

All patients successfully completed the study, and there were no losses to follow-up. As a result, there were no missing values in our dataset, and therefore, imputation methods such as LOCF or multiple imputation were not necessary. The text has been modified to highlight this point.

Despite its limitations and the previously mentioned issues, this study represents a significant step forward in improving nutritional treatment techniques for paediatric patients with non-IgE-mediated gastrointestinal food allergies.

We thank the Reviewer for this comment.

Reviewer 2 Report

Comments and Suggestions for Authors

This is an interesting research article with adequate novelty. Some points should be addressed.

  • The results section could be a bit descreased.
  • The Introduction section is too short and it should be enriched by more data concerning the topic of the research article.
  • The authors should report in the title and the methods section of the abstract that this is a non-randomized, prospective, intervention study.
  • The sentence in lines 94-95 "Other outcomes were to assess whether the mono- or polysensitization and diagnostic delay influence the rate of malnutrition. " is a bit confusing and it should be better explained.
  • The final response rate should be reported.
  • The Table 3 is a bit problematic since in several cells zero cases are reported. The authors should whether is scientifically sound to merge some categories concerning mainly underweight, stunted and wasted participants as well as overweight/obesity. Alternatively, some statement should be reported for the above classification and the corresponding results into the text.
  • The authors should include in the limitations of their study that this is a pilot, non randomized clinical trial with a small sample size, without a control group. Moreover, the nutritional status was assessed by BMI which cannot assessed fat distribution, while a qualified questionnaire could be used to assess malnutrition.
  • English language editing is recommended.
Comments on the Quality of English Language
  • English language editing is recommended.

Author Response

This is an interesting research article with adequate novelty. Some points should be addressed.

The results section could be a bit decreased.

The text has been revised according to this suggestion.

The Introduction section is too short and it should be enriched by more data concerning the topic of the research article.

The text has been revised according to this suggestion.

The authors should report in the title and the methods section of the abstract that this is a non-randomized, prospective, intervention study.

The title and abstract have been modified.

The sentence in lines 94-95 "Other outcomes were to assess whether the mono- or polysensitization and diagnostic delay influence the rate of malnutrition. " is a bit confusing and it should be better explained.

We investigated whether the nutritional status alterations rate was influenced by the presence of single food allergy vs. multiple food allergies, or by the diagnostic delay (expressed as days elapsed between the onset of symptoms and diagnosis). The text has been modified according to this criticism.

The final response rate should be reported.

The final response rate, defined as the rate of patients who showed the resolution of at least one nutritional status alteration subtype (overweight, obesity, underweight, stunting, and wasting) at T12, was 90.3%.

The text has been modified to better highlight this relevant point.

The Table 3 is a bit problematic since in several cells zero cases are reported. The authors should whether is scientifically sound to merge some categories concerning mainly underweight, stunted and wasted participants as well as overweight/obesity. Alternatively, some statement should be reported for the above classification and the corresponding results into the text.

We acknowledge that several cells in Table 3 contain zero cases. However, we have maintained these categories to ensure consistency with established classifications of nutritional status alterations and to align with previous studies in this field. The text has been modified.

The authors should include in the limitations of their study that this is a pilot, non randomized clinical trial with a small sample size, without a control group. Moreover, the nutritional status was assessed by BMI which cannot assessed fat distribution, while a qualified questionnaire could be used to assess malnutrition.

We have added these limitations into the revised version of the text. Moreover, as outlined in the methods section of the manuscript, although we did not assess body composition directly, we did not evaluate malnutrition solely based on BMI values. Instead, by using the Z-scores for all anthropometric parameters, and following the WHO classification, we classified nutritional status alterations as follow: overweight (weight-for-length or BMI >2 and ≤3 Z-score up to 5 years of age; BMI >1 and ≤2 Z-score from 5 to 19 years of age), obesity (weight-for-length or BMI >3 Z-score up to 5 years of age; BMI >2 Z-score from 5 to 19 years of age), moderate underweight (weight-for-age ≤−2 and ≥−3 Z-score), severe underweight (weight-for-age <−3 Z-score), moderate stunting (length/height-for-age ≤−2 and ≥−3 Z-score), severe stunting length/height-for-age <−3 Z-score), moderate wasting (weight-for-length or BMI ≤−2 and ≥−3 Z-score), and severe wasting (weight-for-length or BMI <−3 Z-score).

English language editing is recommended.

The text has been revised according to this criticism.

Reviewer 3 Report

Comments and Suggestions for Authors

There is value in this paper, although I do have some reflections to share with the authors.

Firstly, the outcomes reported at T12 are rather ‘generic (underweight, stunting, wasting, or excess weight), and do not cover potential micronutrient or specific deficiencies. Some additional reflection on this (eg Zink, Iron, or other micronutrient). In the full paper, you use ‘anthropometric parameters’, and this is likely more accurate.

Was there also an assent procedure to the children involved ?

On the stat analysis, why have the other time points not been analysed (taking the non independent character into account).

Not sure if I understand the word choice on ‘alterations’ well, as this - in my opinion – is a synonym for changes, while you rather refer to subclassification on anthropometrics ?

Comments on the Quality of English Language

nothing to add on language

Author Response

There is value in this paper, although I do have some reflections to share with the authors.

Firstly, the outcomes reported at T12 are rather ‘generic (underweight, stunting, wasting, or excess weight), and do not cover potential micronutrient or specific deficiencies. Some additional reflection on this (e.g., zinc, iron, or other micronutrient). In the full paper, you use ‘anthropometric parameters’, and this is likely more accurate.

In the Results section of the manuscript, we reported the outcomes results by dividing the effects of the nutritional intervention on the anthropometric parameters (section 3.2) and nutritional status alterations subtypes (section 3.3) of the WHO classification at T12. Unfortunately, for ethical reasons, we were not able to collect blood samples for measuring biochemical parameters of nutritional status. This limitation has been reported in the text.

Was there also an assent procedure to the children involved?

We collected the written informed consent from the parents/legal guardian of each patient and written assent form of patients from 5 years and older.

On the stat analysis, why have the other time points not been analysed (taking the non independent character into account).

As per study design, we focused on comparing baseline (T0) and the 12-month follow-up (T12), to assess the long-term impact of dietary counseling on nutritional status. While data at intermediate time points (T1 and T6) were collected for clinical purposes, they were not included in the main statistical analysis to avoid potential issues related to multiple comparisons and to ensure the robustness of our findings.

Not sure if I understand the word choice on ‘alterations’ well, as this - in my opinion – is a synonym for changes, while you rather refer to subclassification on anthropometrics?

We referred to nutritional status alterations (or malnutrition subtypes), as reported by the WHO classification, by using the Z-scores for all anthropometric parameters, which are the following: overweight (weight-for-length or BMI >2 and ≤3 Z-score up to 5 years of age; BMI >1 and ≤2 Z-score from 5 to 19 years of age), obesity (weight-for-length or BMI >3 Z-score up to 5 years of age; BMI >2 Z-score from 5 to 19 years of age), moderate underweight (weight-for-age ≤−2 and ≥−3 Z-score), severe underweight (weight-for-age <−3 Z-score), moderate stunting (length/height-for-age ≤−2 and ≥−3 Z-score), severe stunting length/height-for-age <−3 Z-score), moderate wasting (weight-for-length or BMI ≤−2 and ≥−3 Z-score), and severe wasting (weight-for-length or BMI <−3 Z-score).

Round 2

Reviewer 1 Report

Comments and Suggestions for Authors

The manuscript has been revised in accordance with the reviewers' comments, and we agree with the suggested changes. It is now suitable for publication.

Reviewer 2 Report

Comments and Suggestions for Authors

The authors have significantly improved their manuscript.